# Distributed Multi-Robot-Trailer Scheduling Based on Communication between Charging Stations for Robot Being Towed to Recharge

**Yanfang Deng** [1,2]**, Taijun Li** [1,2]**, Mingshan Xie** [3,]***** **and Song Zhang** [3]

1   School of Information and Communication Engineering, Hainan University, Haikou 570228, China
2   Key Laboratory of Internet Information Retrieval of Hainan Province, Haikou 570228, China
3   College of Big Data and Information Engineering, Guizhou University, Guiyang 550025, China
*   Correspondence: msxie@gzu.edu.cn

**Abstract:** It is necessary for robot-trailer to tow robot to the charging station for recharging when the robot fails, or the battery cannot support the robot to the charging station. However, the delay of towing robot is affecting the working efficiency of mobile robot. Based on the task priority of the mobile robot and impact degree on the room after the power failure, the paper proposes a distributed scheduling of robot being towed to recharge for reducing the delay expectation. This work designs a Distributed Three Nodes Service (DTNS) scheduling based on the communication between charging stations. The two-step path-planner based on decision factor and travel path is used in the scheme. Simulations show that the distributed scheduling of this work can well ensure the success communication in the case of low power, and DTNS can well reduce the delay expectation of towing robot to recharge. Compared with First Come First Service (FCFS) scheduling, DTNS reduces the towing delay by 48.71%, 48.83% and 40.45% when there are some robots sending the towing request, and by 58.77%, 39.97% and 41.90% when no robot sends request in the case of 1, 2 and 3 robot trailers in the service space respectively.

**Keywords:** robot-trailer; distributed scheduling model; impact degree

## 1. Introduction

With the recent outbreak of COVID-19, the application of mobile robots has attracted more and more focus, especially the indoor context. Now there are many sale-robots in shopping malls, handling robots in warehouses, production robots in unmanned farms, material handling robots in factories and so on. These robots are all operated automatically. In terms of these robot recharging, current methods typically send the robot to charge when its battery is low, or designate a period of the day within which it always charges. In either case, the generated behaviors can be inflexible and inefficient, not considering task [1].

The task of robot is uncertain. It is possible for robots to still perform tasks when their power is very low. At the same time, since the distance between the mobile robot and the charging station is also uncertain, the power of mobile robot is probably not enough to support it to reach the charging station for recharging. If the battery discharges totally, it is also very possible that the robot cannot reach the charging station automatically. Then human intervention is required to manually move the robot to the charging station [1]. However, the traditional method is to use manpower to tow the robot to the charging station. This issue requires the robot-trailer to tow the robot that cannot reach the charging station for recharging. It can reduce labor costs and reduce the impact of robot power failure. The towing scheduling of robot after power failure is particularly important, and its delay directly affects the efficiency of the whole robot system. The optimization of towing scheduling is a serious issue needed to be solved.

In terms of trailer towing scheduling, there are many studies on trailers in ports and airports. Zheng et al. [2] proposed Greedy Randomized Adaptive Search Procedure (GPASP) scheduling algorithm for trailer scheduling in cold chain logistics. Zhou et al. [3] analyzed the operation rules of the airport trailer, combined with the service process and the characteristics of the trailer, established a multi-objective trailer emergency scheduling model with rolling window, and introduced the simulated annealing algorithm into the genetic algorithm to solve the scheduling model. The model can more effectively solve the airport trailer emergency scheduling problem in the case of flight delay and alleviate the pressure of flight delay in busy airports. The authors of [4] analyzed the vehicle scheduling problem in the factory environment, which involves various types of vehicles and different tasks, as well as the time window problem of vehicle scheduling. These towing scheduling are completed by workers operating trailers.

With the development of automatic systems, it will become automatic and unmanned to tow the robot to charging station. Since the concept of robot-trailer was put forward in 2007, people have been studying robot-trailer to realize unmanned scheduling. Hodo et al. [5] solved the problem of controlling trailer lateral position. The control device consists of a Global Positioning System (GPS) receiver and a suspension sensor to measure the angle between the robot (tractor) and the trailer. In the same year, to solve the influence of sensor position and measurement error on the path tracking control of robot-trailer, they [6] designed a state feedback controller to make the trailer locate along the path composed of straight line and arc, and adopted the combined system of GPS and Inertial Measurement Unit (IMU) to measure the position and direction of robot person or trailer (but not both), so that the robot-trailer is less sensitive to errors in the hitch angle measurement. Based on the local obstacle detection algorithm of fast stereo vision processing steps, combined with dynamic path planning, Tsiogas et al. [7] designed a path planner for the robot-trailer to avoid obstacles in real time, used the global path planning algorithm to complete the robot-trailer coverage and comprehensive search of the operation area, and used the local dynamic path planning algorithm to realize dynamic obstacle avoidance. Yu et al. [8] equipped the robot-trailer with sensing equipment to perform monitoring tasks between multiple waypoints of interest. The designed shortest path planning algorithm has good performance in both low waypoint density and high waypoint density.

In this paper, the scheduling of robot-trailer towing power-off robot is considered. We have made some endeavor to solve it. This issue poses three challenges:

(1) The position of the robot sending the recharging request to be towed is uncertain. How to ensure that the whole charging system can receive the request under the condition of very low battery?

(2) The time when the robot sends the recharging request to be towed is uncertain and random. How to ensure that robots with important tasks are towed to charge as soon as possible?

(3) After the robot loses power in the room, it will affect people's activities or the running track of other robots. Especially for rooms with large passenger flow, the impact degree increases with passenger flow. How to ensure that the impact on rooms with large traffic is limited?

Inspired by the above challenges, based on the communication between charging stations, we study the optimization of robot towed to recharge after the power is exhausted. Combining multi-robots and robot-trailers, the distributed scheduling model of robot towed to recharge is proposed. The path-planner of the robot-trailer is designed, and the mean charging delay of the whole robot service system is used to evaluate the towing charging scheme. The contributions of this work are outlined as follows:

(1) Considering the impact on the room where the robot is located and the weight of the task of robot itself, the scheduling optimization of robot towed to recharge is proposed after robot power is exhausted.

(2)    A Distributed Three Node Service (DTNS) scheduling model of robot-trailer based on charging station communication is proposed. The cooperative scheduling of robot-trailer between charging stations is realized.

(3)    A two-step path-planner based on decision factor and travel path is proposed.

The rest of this paper is organized as follows. Section 2 presents related works on scheduling. Section 3 describes the indoor mobile service system and constructs the model. Section 4 introduces the distributed three node service scheduling model and the two-step path-planner. Section 5 demonstrates the simulations to validate the accuracy and the efficiency of the method and conducts correlation analysis. Finally, conclusions are drawn in Section 6.

## 2. Related Work

The construction and optimization of scheduling model is the core issue of multi-robot-trailers towing robots to recharge. Scheduling models are divided into two types: one is the centralized, and the other is the distributed.

So far, there has been a lot of research work on centralized scheduling. Zhang et al. [9] established the centralized pick-up truck scheduling model to solve the pick-up problem of trailer. They adopted the improved C-W (Clarke Wright) algorithm, and successfully found the optimal logistics distribution path of 9 customer cases, according to the fuel consumption equation of different load capacity, taking reducing fuel cost, traffic cost, loading and unloading cost and delay penalty cost as the minimum objective function. A centralized scheduling model with the optimization goal of minimizing the cost and load balance of industrial robots was presented in [10] to complete the coordinated work of industrial robots in different manufacturing centers. The authors used artificial immune algorithm to obtain the best scheduling process. In order to solve the problem that transportation robots and picking robots cooperate to execute multi station order fulfillment tasks in the logistics environment, with the goal of minimizing the completion time, Wang et al. [11] designed a sequence based on task model and solved it with the coupled heuristics algorithm. When the number of pickup robots is relatively small and the working area is large, the effect of the coupled heuristic algorithm is obviously better than the decoupled heuristics algorithm. The authors of [12] designed a new economic dispatching optimization model of electric vehicle, and adopted the multi-level optimization V2G dispatching method to ensure the smooth operation and control from V2G control center to electric vehicle users. In this work, it is realized to minimize the operation cost of regional Vehicle to Gird (V2G) system and solve the load fluctuation problem of electric vehicle charging to the power grid. The authors in [13] established an integer programming model with the goal of minimizing the number of ferry vehicles required in peak hours or a certain period to the optimize the scheduling of ferry vehicles in flight ground support services. By introducing virtual flight, a ferry vehicle sharing network is constructed, and the model is transformed into the network maximum flow problem. The effectiveness and accuracy of the method are verified by using the flight data of Beijing Capital International Airport. In [14], the new bi-objective mixed-integer linear programming model is proposed to address the vehicle routing problem with cross-docking (VRPCD) that considers truck scheduling, splitting pickup and delivery orders with time-windows at supplier and retailer locations, while optimizing two conflicting objectives (i.e., cost efficiency and responsiveness). The multi-objective meta-heuristic evolutionary algorithm is proposed to minimize the total operational cost and the sum of the maximum earliness and tardiness, whose effectiveness is better than Non-dominated Sorting Genetic Algorithm (NSGA-II) and Pareto Archiving Evolutionary Strategy (PAES).

Many application scenarios also require distributed scheduling. In the job scheduling and resource allocation of high-end systems, to improve the global throughput, Hafaiedh et al. [15] proposed a fully distributed formal model to describe different job scheduling algorithms. A set of parallel schedulers are defined, which communicate with each other to achieve a given load balancing strategy. The model has certain applicability in grid environment. For

multi-robot systems, Ferreira [16] proposed a distributed task allocation and scheduling algorithm, which is used for tasks closely coupled with time and priority constraints of different robots. The distributed meta heuristic algorithm based on evolutionary computing is used to achieve fast and near optimal allocation. This method has better computing speed and scalability without losing optimality. A distributed algorithm that enables a group of robots to jointly optimize the parameters of the deep neural network model while communicating through the mesh network is studied in [17]. In each iteration, each robot approximately optimizes an augmented Lagrange function, then passes the obtained weight to its neighbors, updates the bivariate, and repeats. For convex objective function, this method can get a global optimal solution. The authors of [18] proposed a distributed optimization algorithm for scheduling the energy consumption of multiple smart homes with distributed energy resources. Numerical results demonstrated that the proposed distributed algorithm achieves almost equivalent performance to that of the centralized optimization algorithm in terms of consumer's electricity cost and comfort level. The authors of [19] proposed a distributed method to navigate robot formation in two-dimensional and three-dimensional environments with static and dynamic obstacles, assuming that the communication and visibility radius of these robots are reduced and share information with their neighbors. They calculated (a) the converge hull of the robot positions and (b) the largest converge region within free space through distributed consensus. The approach is efficient and scalable with the number of robots and performs well in simulations with up to sixteen quadrotors. Kashipaz et al. [20] studied the distributed formation control of multi robot systems with limited communication resources and unknown process measurement noise, and proposed an event triggered communication scheme based on extended Kalman filter estimation to improve the communication rate. For multi-agent systems, the authors of [21] designed a distributed dynamic event triggering framework for multi-agent systems to reduce the communication between leaders and followers. Luis et al. [22] proposed a Distributed Model Predictive Control (DMPC) algorithm for multi-agent offline trajectory generation. By predicting the future state and sharing this information with their neighbors, agents can detect and avoid collisions while moving towards their goals. They [23] also apply DMPC algorithm to generate the trajectories of multiple robots in real time. Compared with the fully studied Buffered Voronoi Cells (BVC) approach, the proposed collision avoidance method can reduce the travel time required to complete multi-agent point-to-point conversion by about 50%. Filho et al. [24] solved the motion planning problem of a group of wheeled mobile robots, and proposed a distributed mathematical planning method. Perception, trajectory planning and execution are staggered, and can be executed independently on each robot, which well realizes collision avoidance. The authors of [25] proposed a method for uniformly distributed localization and semantic mapping of multiple robots in unknown environment. Compared with using only local information to maintain mixed belief, the classification and localization accuracy are improved. Sung et al. [26] studied the problem of tracking multiple moving targets using a group of mobile robots. Aiming at maximizing the number of tracked targets, they designed a distributed algorithm based on greedy algorithm and local algorithm to realize the trade-off between tracking quality and communication time. Ferreira et al. [27] proposed a distributed multi-stage optimization method for planning complex tasks of heterogeneous multi robot teams.

Distributed scheduling is conducive to the stability of the whole system and improve the robustness of the system to complete tasks. Distributed scheduling provides a research direction for robot-trailer scheduling.

## 3. Problem Statement

To address the issue of indoor robot-trailer scheduling, the authors set that there are $m$ rooms in the service indoor space $\Omega$ of the robot. $\Omega = \bigcup\limits_{j=1}^{m} Z_j$, where $Z_m$ represents the $m$th room. The area of service space $\Omega$ is $s_\Omega$. The flow of people in each room is different, so the impact of the robot on the room after the power runs out is also different. After the robot runs

out in the room, the impact on the room $Z_j$ is set to $w_{Z_j}$. $W_\Omega = \{w_{Z_1}, w_{Z_2}, \cdots, w_{Z_m}\}$, where $W_\Omega$ is a set of $w_{Z_j}$. The more frequent the flow of people in the room where the robot is located, the higher the impact on experience of users and work efficiency of other robots in the room after the power of the robot is exhausted, because the robot parking in the room will affect people walking or other robots operating in the room. $w_{Z_j}$ can be calculated by synthesizing the time spent by people in the room per unit time $\tau$, as shown in Formula (1).

$$w_{Z_j} = \frac{\sum\limits_{i=1}^{|\Gamma_{Z_j}|} t_i^{Z_j}}{\sum\limits_{j=1}^{m} \sum\limits_{i=1}^{|\Gamma_{Z_j}|} t_i^{Z_j}} \tag{1}$$

where $\left|\Gamma_{Z_j}\right|$ is the number of people or robots in the room $Z_j$ per unit time, $t_i^{Z_j}$ represents the time spent by the $i$th person or robot in the room $Z_j$. $h$ robots are deployed in the service space. $R = \{r_1, r_2, \cdots, r_h\}$, where $r_h$ is the $h$th robot, $R$ is a set of robots. Each robot has its own responsible task, so the importance of the task is different, $W_R = \{w_{r_1}, w_{r_2}, \cdots, w_{r_n}\}$, where $w_{r_i}$ represents the impact on the whole robot system after the power of the robot $r_i$ is exhausted, $W_R$ is a set of weights of robots. The larger $w_{r_i}$, the more important the task of robot $r_i$ is, and the greater the impact on the whole system after its power is exhausted. $\mu$ robot-trailers, which are used to tow the robot to recharge when the power of the robot is exhausted, are deployed in the whole service space. $L = \{l_1, l_2, \cdots, l_\mu\}$, where $l_\mu$ represents $\mu$th robot-trailer, $L$ is a set of robot-trailers. $n$ fixed intelligent charging stations have been deployed, $S = \{s_1, s_2, \cdots, s_n\}$ where $s_n$ represents $n$th charging station, $S$ is a set of charging stations. The working mode of the robot-trailer is shown in Figure 1, where each square represents a room. Robot-trailers can communicate with each other between robots and charging stations, among charging stations, and between robot trailers and charging stations. Short distance communication technologies such as Wi-Fi and ZigBee are usually used for indoor communication. Each robot performs its own task according to the corresponding trajectory in the offline model.

In the indoor application scenarios, if the current power of the robot cannot drive to the nearest charging station, the robot sends the request to be towed to the station. The robot can monitor its power level at any time [28]. Let the threshold of the electric quantity of the robot being towed to charge be $g_{th}$ and the current electric quantity of the robot be $g_{real}$. If $g_{real} \leq g_{th}$, then the robot sends the request to be towed to the nearest charging station for recharging, at the same time sends the current position $r_i(x_{r_i}, y_{r_i})$ of the robot $r_i$ to the nearest charging station. The robot-trailer will tow the robot to a fixed charging station for recharging.

Due to the different tasks performed by robots and their different energy consumption rates, the time when each robot sends the recharging request is also different. $\theta_{r_i}$ represents energy consumption rates of the robot $r_i$. The transmission of recharging request is realized through wireless technologies such as point-to-point communication [29]. The transmission time of the charging request is negligible relative to the charging time. The charging delay of each robot is defined as the time difference from the time when the robot sends the towing charging request to the time when the robot gets the charging service. The delay time $t_{tra}(r_i)$ of $r_i$ robot waiting for charging service is defined as:

$$t_{tra}(r_i) = t_{now}(r_i) - t_{request}(r_i) \tag{2}$$

where $t_{now}(r_i)$ is the time when the robot $r_i$ is towed to the charging station by the robot-trailer. $t_{request}(r_i)$ is the time when the robot $r_i$ sends request to be towed for recharging.

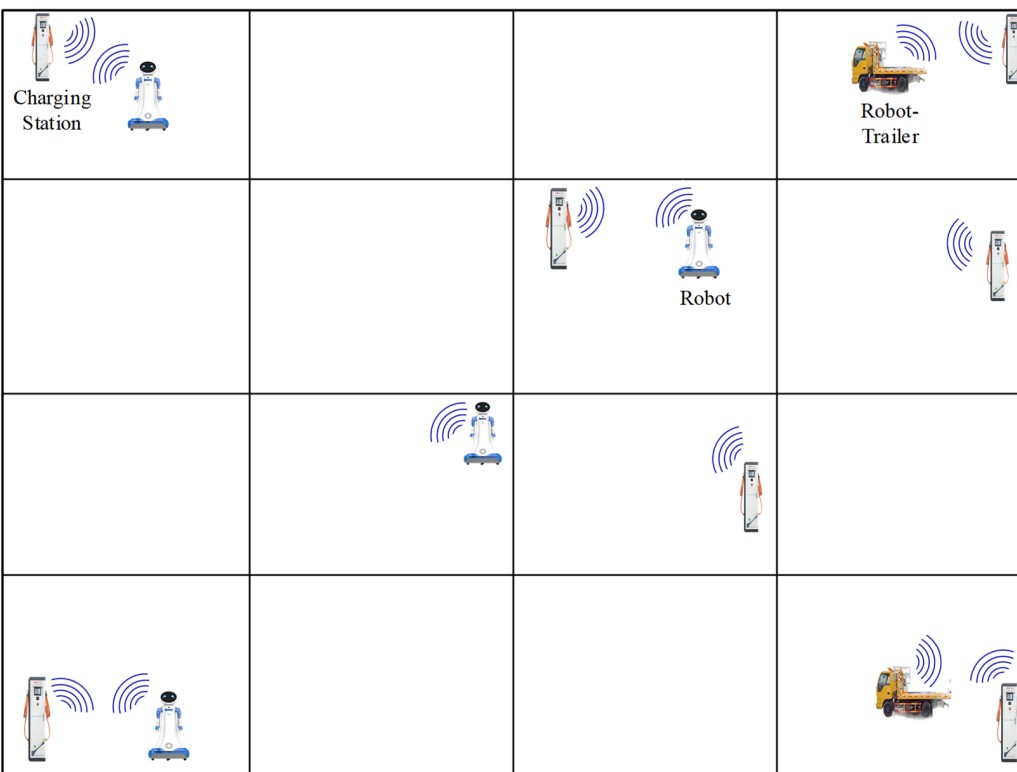

**Figure 1.** Working diagram of indoor robot-trailer.

This paper sets a robot trailer to tow only one robot. After the robot-trailer transports the robot to the charging station, it starts from the charging station and moves to the next robot position that sends a charging request. Formula (2) can be transformed into Formula (3).

$$t_{tra}(r_i) = t_{sedu}(r_i) + \frac{d_{r_i \to s_i}}{v_l} \tag{3}$$

where $t_{sedu}(r_i)$ represents scheduling waiting time caused by scheduling algorithm of the robot $r_i$. $v_l$ represents the speed of robot-trailer. Let the speed of each robot-trailer be the same and move at a uniform speed. $d_{r_i \to s_i}$ is the distance traveled by the robot-trailer to tow the robot $r_i$ to the charging station $s_i$ for charging.

For rooms with large traffic, the user's satisfaction and the working efficiency of other robots will be affected by the waiting time of robots in such rooms. For robots responsible for important tasks, they should be charged as soon as possible after the power is exhausted, to reduce the loss.

To evaluate the execution efficiency of the robot-trailer in the robot service space, the delay expectation of the whole robot group in unit time is used as the key measure in this paper. On average, $\gamma$ robots send out towing charging requests in unit time, $\gamma \leq h$. The delay expectation of robot group is shown in Formula (4).

$$E = \sum_{i=1}^{\gamma} t_{delay}(r_i) \times \left( w_{z_j} \big|_{r_i \in Z_j} + w_{r_i} \right) \tag{4}$$

where $w_{z_j} \big|_{r_i \in Z_j}$ is the impact on the room where the robot sends the request to be towed for recharging.

The goal of robot-trailer scheduling optimization is to solve the load balance problem of robot-trailer and reduce the impact caused by the robot after its power is exhausted. After the robot runs out of power or breaks down, if the scheduling planning is not carried out, the waiting time of the robot performing important tasks and the robot in the important



room will be too long, resulting in the low user experience of the robot and the reduced efficiency of other robots. In order to achieve the minimum delay expectation of the robot service system, the scheduling optimization model of the robot- trailer is defined as:

$$P^*(\Omega, W_R, W_\Omega, S, R) = \text{argmin}E \tag{5}$$

## 4. Model Solution and Analysis

To ensure the overall charging efficiency of the robot service system, more than one robot-trailer is required. The distributed scheduling model for the recharging of robot towed to charging station by robot-trailer is proposed to optimize the efficiency of recharging after the robot runs out of power. What we need to solve is to allocate which robot-trailer to tow robot to charge when power of robot is exhausted, that is, the allocation scheme about the robot-trailer and the robot.

### 4.1. Distributed Communication

When the robot sends the request, the energy is not enough. It is difficult to communicate with the remote station receiving request. $p_{suc}$ represents the success rate of communication between the robot sending request to be towed to recharge and the station receiving request, then it can be calculated by Formula (6).

$$p_{suc} = p(d_{real} \leq d_{th}) \tag{6}$$

where $d_{real}$ represents the physical distance the robot sending the request to be towed for recharging and the station receiving request. $d_{th}$ represents the communication distance threshold causing communication failure, that is, the distance that the radio wave sent by the robot can transmit. If $d_{real} > d_{th}$, then robot cannot communicate with the service station, and the towing recharging request fails. The success rate of communication is essentially the probability that $d_{real}$ is less than or equal to $d_{th}$.

The distributed robot-trailer scheduling model requires that the robot-trailer can communicate with the charging station. The robot-trailer can communicate with the charging station through Bluetooth communication technology. The robot-trailer receives the information from the charging station near it. Charging stations can also communicate with each other through Wi-Fi or ZigBee technology. The distributed robot-trailer scheduling in this paper is to improve the success rate of the robot requesting service, that is, to improve $p_{suc}$, and adopts the way that the robot sends the request to be towed for recharging to the charging station.

### 4.2. Distributed Three Nodes Service Scheduling

When the robot sends the request to be towed for recharging, the nearest charging station will store the position of the robot to be towed in dataset $\Upsilon$ after receiving the request, $\Upsilon = \{r_k(x_{r_k}, y_{r_k}), \gamma | \zeta_{r_k} = 0, k \in (1, 2 \cdots h)\}$, where $\zeta_{r_k}$ represents the state of robot, which $\zeta_{r_k} = 0$ represents robot $r_k$ is waiting to be towed for recharging and $\zeta_{r_k} = 0$ represents robot $r_k$ is working. $\gamma$ represents update timestamp of dataset $\Upsilon$.

Set the locations of the idle charging station that can be used to be stored as $\Psi = \{s_i(x_{s_i}, y_{s_i}), \psi | q_{s_i} = 0, i \in (1, 2 \cdots q)\}$, where $q_{s_i}$ represents the state of charging station, which $q_{s_i} = 0$ represents the charging station $s_i$ is idle and available and $q_{s_i} = 1$ represents the charging station $s_i$ is occupied. $\psi$ represents update timestamp of the dataset $\Psi$.

When the robot-trailer comes to the charging station, it receives $\Upsilon$ and $\Psi$. Based on the stored map of service space, the robot-trailer plans the robot to be towed and the destination where it will tow the robot to recharge and then deletes the planned robots and charging stations in $\Upsilon$ and $\Psi$. The robot-trailer sends the updated $\Upsilon$ and $\Psi$ to other charging stations. The updated $\Upsilon$ and $\Psi$ are transmitted and shared between charging stations. The process of sending and receiving $\Upsilon$ and $\Psi$ from the charging station is shown in Figure 2.

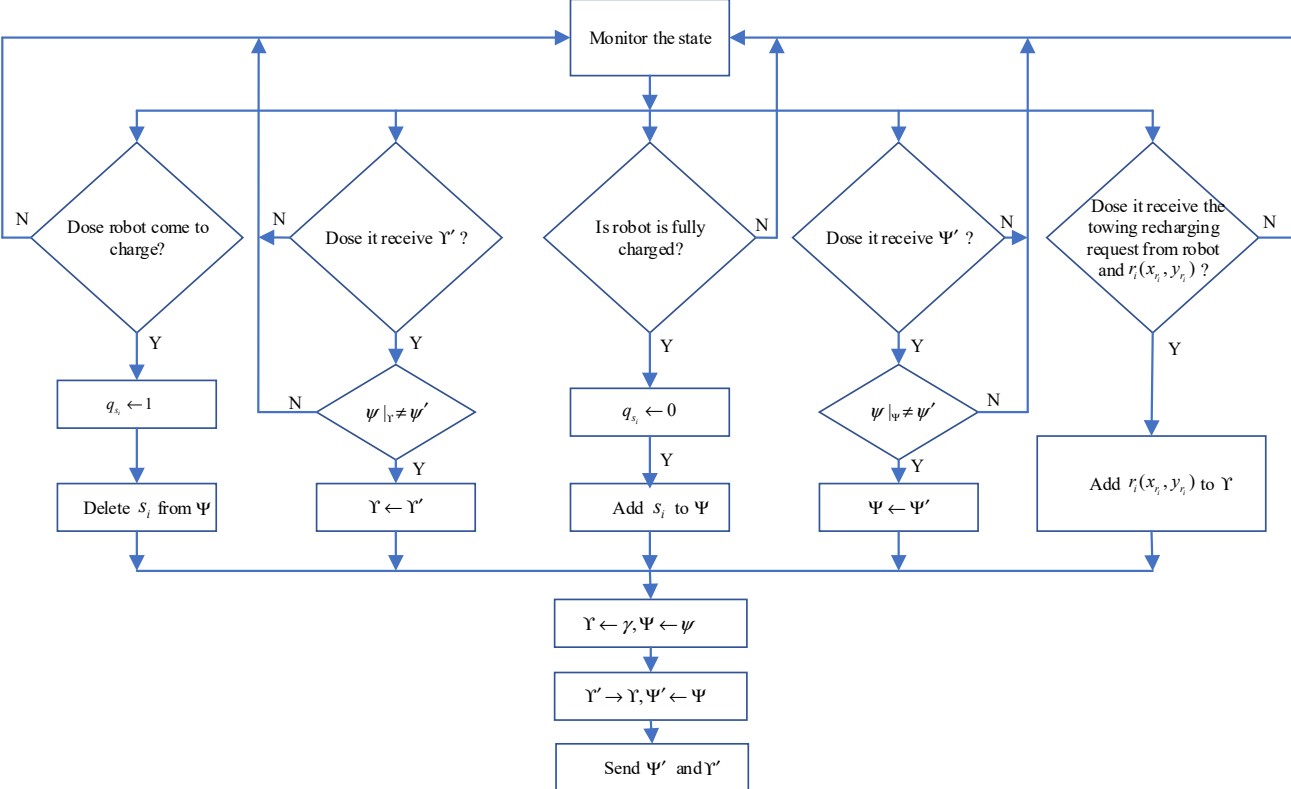

**Figure 2.** Flow diagram of charging station receiving and sending $\Upsilon$ and $\Psi$.

The charging station always monitors its various states. If a robot comes to charge at the charging station, that is, the charging station is occupied, the state $q_{s_i}$ is changed to 1 and deleted from the dataset $\Psi$. When the charging station receives the position dataset $\Upsilon'$ of the robot to be towed from other charging stations or robot trailers, it first determines whether the received time stamp $\psi$ is equal to its own time stamp $\psi|_{\Upsilon}$. If not, it indicates that the received dataset is updated. The charging station overwrites the original dataset $\Upsilon$ with the received dataset $\Upsilon'$. When the charging station receives the idle charging station location dataset $\Psi'$ sent by other charging stations or robot trailers, it first judge whether the received time stamp $\psi$ is equal to its own time stamp $\psi|_{\Psi}$. If not, it indicates that the received dataset is updated, and the charging station will overwrite the original $\Psi$ with the received $\Psi'$. If a robot leaves with full charge, that is, the charging station is idle and available, the state $q_{s_i}$ of the charging station is changed to 0, and added to $\Psi$. When the charging station receives the towing recharging request from the robot, it will store the robot position $r_i(x_{r_i}, y_{r_i})$ in $\Upsilon$. Finally, the charging station stores the modified timestamp in $\Upsilon$ and $\Psi$, and copies $\Upsilon$ and $\Psi$ to other charging stations and the nearest robot-trailers.

When the robot-trailer arrives at the charging station to unload the robot, it plans its own task path, The task link list of the robot-trailer at the charging station $s_k$ is set to $X_t = \{s_k(x_k, y_k) \rightarrow r_i(x_i, y_i) \rightarrow s_j(x_j, y_j)\}$, where $t$ is the timestamp when robot-trailer planning task, $a \rightarrow b$ represents the shortest path from position $a$ to position $b$, which can be solved by path planning algorithms such as $A*$ algorithm or particle swarm optimization algorithm. $X_t$ stores the coordinates of the charging station and the requesting robot, with the position of the charging station nearest to the robot-trailer as the head node and the position of the charging station as the tail node. $X_t$ is set to have only three nodes, as shown in Figure 3.

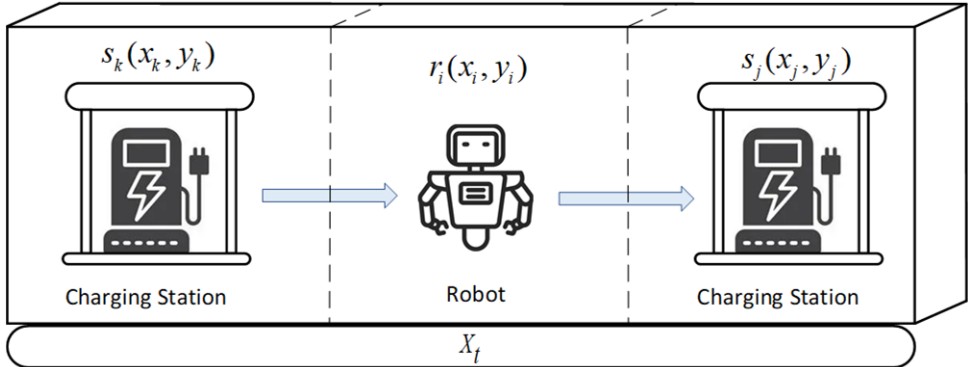

**Figure 3.** Three nodes task link list of robot-trailer.

Each robot-trailer carries out the towing task according to its corresponding $X_t$, and transports the robot with exhausted power according to the order of nodes of $X_t$. $X_t[0]$, $X_t[1]$ and $X_t[2]$ represents the first, the second and the third node in $X_t$. The robot-trailers go back and forth between the charging station and the requesting robot.

When the robot-trailer arrives at the charging station, it carries out towing task planning. Robot-trailers plan their own task link list, which is a distributed towing recharging scheduling. If the centralized scheduling is adopted or the number of nodes in the task link list is greater than three, it will cause repeated planning. Because if the new coordinates come in $\Upsilon$ and $\Psi$, after the robot-trailer has been planned, in order to reduce the waiting expectation, robot-trailer needs to re-plan the task list to schedule these high priority task nodes. The distributed towing scheduling of this work divides all towing tasks into parts. The DTNS scheduling enables the robot-trailer to receive the new towing charging request and the newly released charging station at any time by using the wireless network when performing the whole towing task. If the task link list of the robot-trailer is larger than three, it will make it impossible to plan the path included the new high weight robot that sends a towing recharging request or the high priority charging station that is released before the task is completed.

*4.3. Two-Step Path-Planner*

To achieve the minimum delay expectation, the path planning of robot-trailer scheduling is divided into two steps.

Step 1: Robot-trailer $l_i$ selects the robot position with the smallest decision factor from the charging station to requesting robot from the dataset $\Upsilon$;

Step 2: Robot-trailer $l_i$ selects the position of the charging station with the smallest distance from requesting robot from $\Psi$, and then plan the path to tow the robot to recharge.

There are many factors involved in selecting the robot that sent the request to be towed to recharge, such as the importance of the robot's task and the impact of the robot's room on human activities. $\eta_{X_t[0]\to r_i}$ is used to represent the decision factor from the charging station to the requesting robot, that is, the tendency of choosing which requesting robot to start from the charging station.

When solving $\eta_{X_t[0]\to r_i}$, the distance should be normalized first. The distance from the first node in task link list of robot-trailer to the requested position of the robot is normalized and calculated by Formula (7):

$$d_{X_t[0]\to r_i}^{norm} = \frac{d_{X_t[0]\to r_i} - d_{X_t[0]\to r_i}^{\min}}{d_{X_t[0]\to r_i}^{\max} - d_{X_t[0]\to r_i}^{\min}}, r_i \in \Upsilon \tag{7}$$

where $d_{X_t[0]\to r_i}$ represents shortest distance from the first node in task link list of robot-trailer to the requested position of the robot $r_i$, which is calculated by $A*$ algorithm. $d_{X_t[0]\to r_i}^{\max}$ represents maximum value of the shortest path length from the first node in $\Upsilon$ to

all requesting robots. $d^{\min}_{X_t[0] \to r_i}$ represents minimum value of the shortest path length from the first node in task link list of robot-trailer to all requesting robots. $d^{norm}_{X_t[0] \to r_i}$ represents the normalized value of the distance from the first node in task list of robot-trailer to position of the requesting robot, a dimensionless value. The value range of $d^{norm}_{X_t[0] \to r_i}$ is between 0 and 1.

The decision factor from the charging station to the robot is calculated by Formula (8):

$$\eta_{X_t[0] \to r_i} = \alpha \times d^{norm}_{X_t[0] \to r_i} + \beta \times (1 - w_{r_i}) + \phi \times (1 - w_{Z_g|r_i \in Z_g}), i = 1, 2 \cdots h, g = 1, 2 \cdots m \tag{8}$$

where $\alpha$, $\beta$ and $\phi$ are adjustment coefficients, adjusting the proportion of the three indicators. Then the second node of the planning task link list is the position of the towing recharging request robot with the smallest decision factor, as shown in Formula (9).

$$\begin{aligned} X_t[1] &= \operatorname{argmin}\eta_{X_t[0] \to r_i} \\ &s.t.r_i \in \Upsilon \end{aligned} \tag{9}$$

The third node of the task list is the position of the charging station with the smallest travel distance from the charging request robot, as shown in Formula (10).

$$\begin{aligned} X_t[2] &= \operatorname{argmin}d_{X_t[1] \to s_j} \\ &s.t.s_j \in \Psi \end{aligned} \tag{10}$$

where $d_{X_t[1] \to s_j}$ represents the shortest distance from the second node of the task link list to the idle charging station.

The two-step path planning process of robot-trailer proposed by us is shown in Figure 4. At the initial time of path planning of the robot-trailer, it stores the position of the idle charging station near it in the first node of the task planning list, that is, $X_t[0] \leftarrow s_k(x_{s_k}, y_{s_k})$, and then deletes the position of the charging station from $\Psi$. If the path planning has been carried out, the third node in the task link list of the previous planning will be stored in the first node of the task planning list at time $t$, that is, $X_t[0] \leftarrow X_{t-1}[2]$. Calculate $d^{\max}_{X_t[0] \to r_i}$ and $d^{\min}_{X_t[0] \to r_i}$. Judge whether $r_i$ is in $Z_j$, that is, find the room where the robot is located. Then select the importance value $w_{Z_j}$ of the room $Z_j$ where the robot is located from $W_\Omega$. The decision factor $\eta_{X_t[0] \to r_i}$ from the first node of the task planning list to each charging robot is obtained by Formula (8), and the position of the requesting robot corresponding to the minimum decision factor is stored in the second node of the task planning list. And delete the selected position of requesting robot from dataset $\Upsilon$. Calculate the travel from the second node in the task planning list to each charging station, i.e., $d_{X_t[1] \to s_j}$, and store the charging station position corresponding to the smallest travel in the third node of the list. The selected position of charging station is deleted from dataset $\Psi$. The modified time stamp is stored in dataset $\Upsilon$ and dataset $\Psi$, which are sent to the charging station, and finally output $X_t$. The robot-trailer performs the towing and charging task according to $X_t$.

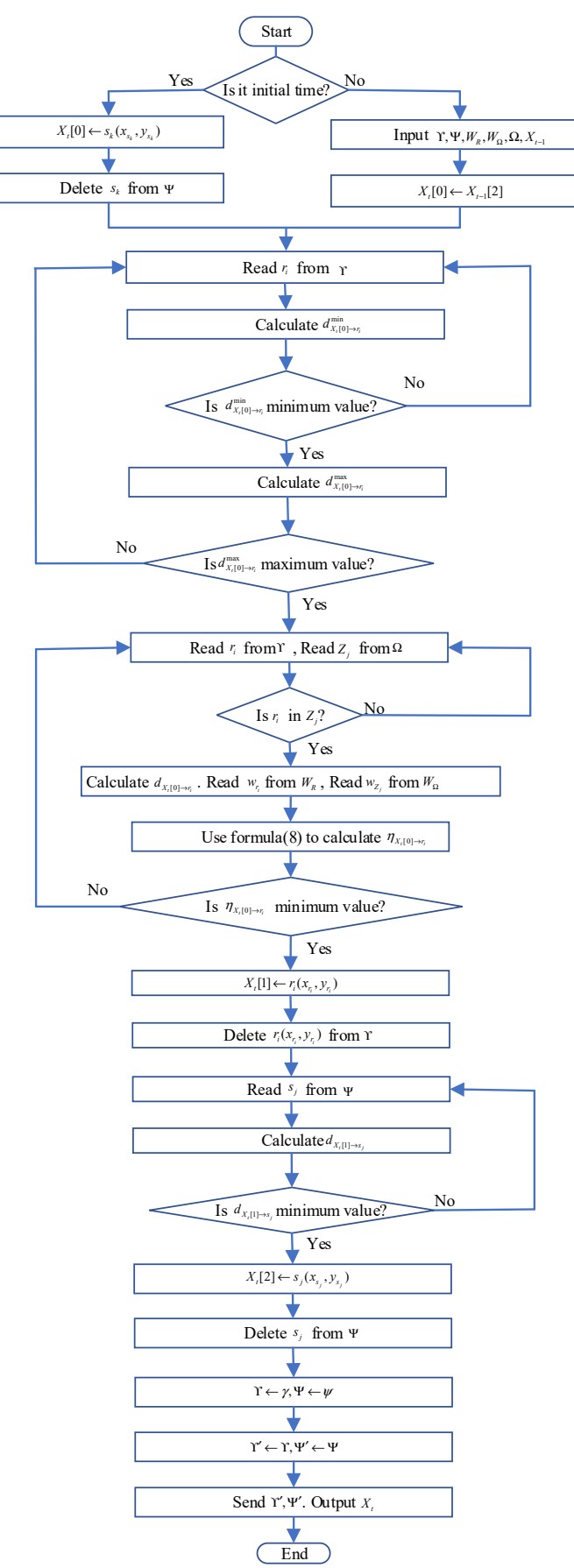

**Figure 4.** Tow-step path planning process of robot-trailer.

## 5. Simulation and Analysis

In this section we simulated the towing recharging optimization scheme. We use MATLAB 2020a tool for simulation verification. It is possible and independent for robots to send requests to be towed for recharging in each room. The position of the robot, charging station and robot-trailer adopts two-dimensional Cartesian coordinate system, with the lower left corner of the space as the coordinate origin.

### 5.1. Simulation 1

The model of this work adopts that the robot sends the request to be towed for recharging to the charging station, and the charging station can communicate with the robot-trailer. It plans its own path to tow the robot to recharge. The robot sends the request to be towed to recharge when the energy is low. If the distance between the requesting robot and the receiving service station exceeds the communication distance threshold, the request will fail. We use the success rate of communication $p_{suc}$ to measure the superiority of towing recharging request communication. It is calculated by Formula (11).

$$p_{suc} = 1 - \frac{N_{lose}}{N_{test}} \tag{11}$$

where $N_{lose}$ represents number of communication failures, $N_{test}$ is total number of requests.

We have verified three modes of robot charging request: The first is that the robot sends a towing request to the robot-trailer; The second is that the robot sends a towing request to the service center; The third is that the robot sends a towing request to the charging station. The distributed scheduling model of this work adopts the third mode. The verification is set in the service space with a fixed area of 40 m × 40 m. The communication threshold changes from 5 m to 50 m. The location of the service center is set to (5 m, 5 m). In this test, the robot made 1000 towing recharging requests. We have discussed two situations in which the charging station is deployed in the service space using fixed location and random uniform distribution. The fixed deployment position of the charging station is assumed to be (10, 10), (20, 20), (30, 10), (10, 30), (30, 30). We compared the variations of communication success rates with the communication threshold when charging stations are deployed in these two modes, as shown in Figures 5 and 6.

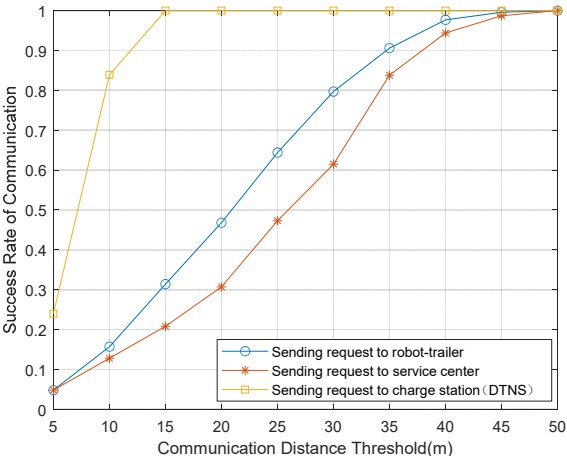

**Figure 5.** Comparison of success rates of communication when the charging stations are deployed in a fixed location.

Figure 5 presents the variations of the success rates of communication and the communication distance threshold when the charging stations are fixed. When the communication distance threshold is 15, the communication success rate of sending the charging request to the charging station (DNTS) reaches 1. When the communication distance threshold is 50, the communication success rates of sending the charging request to the trailer and

service center reach 1. Figure 6 shows the variations of the success rates of communication and the communication distance threshold when the charging stations are randomly distributed in the service space. When the communication distance threshold is 25, the communication success rate of sending the charging request to the charging station (DNTS) reaches 1. When the communication distance threshold is 50, the communication success rates of sending the charging request to the trailer and service center reach 1. It can be seen from Figures 5 and 6 that with the increase of the communication distance threshold, the communication success rate also increases. When the charging station adopts fixed location deployment and random uniform distribution deployment, the communication success rate of sending charging requests to the charging station is higher than that to the robot-trailer, and the success rate of sending charging requests to the robot-trailer is higher than that to the service center. Since both the robot- trailer and the robot are moving, it is more likely that the distance between the request robot and the trailer exceeds the communication threshold. The robot communicates with the service center, which depends heavily on the service center, so the robustness of the system is not high. The model of this work has obvious advantages over sending charging requests to robot-trailers and service centers.

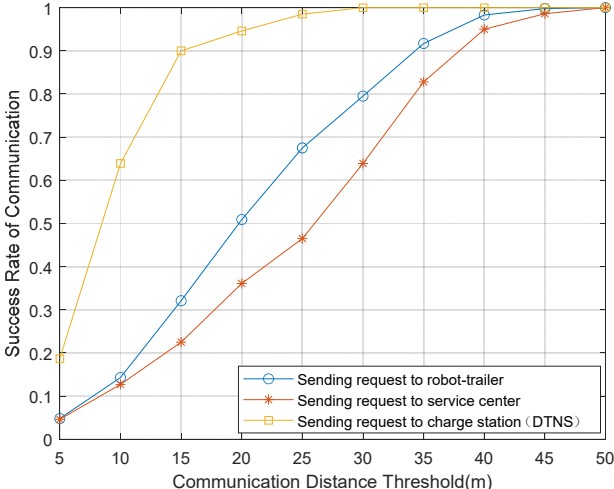

**Figure 6.** Comparison of success rates of communication when the locations of charging stations are randomly distributed in the service space.

*5.2. Simulation 2*

This simulation is to verify the superiority of scheduling efficiency. We compare the delay expectation and delay time between the DTNS scheduling algorithm and the FCFS [30] scheduling algorithm. The service space of this experiment is set in a space of 40 m × 40 m, as shown in Figure 7.

In Figure 7, the black square ■ represents the deployed charging station, the black blob ● represents the position of the robot to be transported for charging, bracketed number (·) represents the door and its number, and the coordinates of each door are shown in Table 1.

**Table 1.** Coordinates of each door.

| Index of Door | Coordinates | Index of Door | Coordinates | Index of Door | Coordinates | Index of Door | Coordinates |
|---|---|---|---|---|---|---|---|
| (01) | (7, 10) | (06) | (30, 2) | (11) | (20, 22) | (16) | (20, 39) |
| (02) | (10, 2) | (07) | (38, 10) | (12) | (10, 28) | (17) | (30, 39) |
| (03) | (20, 2) | (08) | (38, 20) | (13) | (8, 20) | (18) | (38, 30) |
| (04) | (17, 10) | (09) | (10, 12) | (14) | (12, 20) | (19) | (27, 30) |
| (05) | (28, 10) | (10) | (28, 20) | (15) | (10, 39) | (20) | (8, 30) |

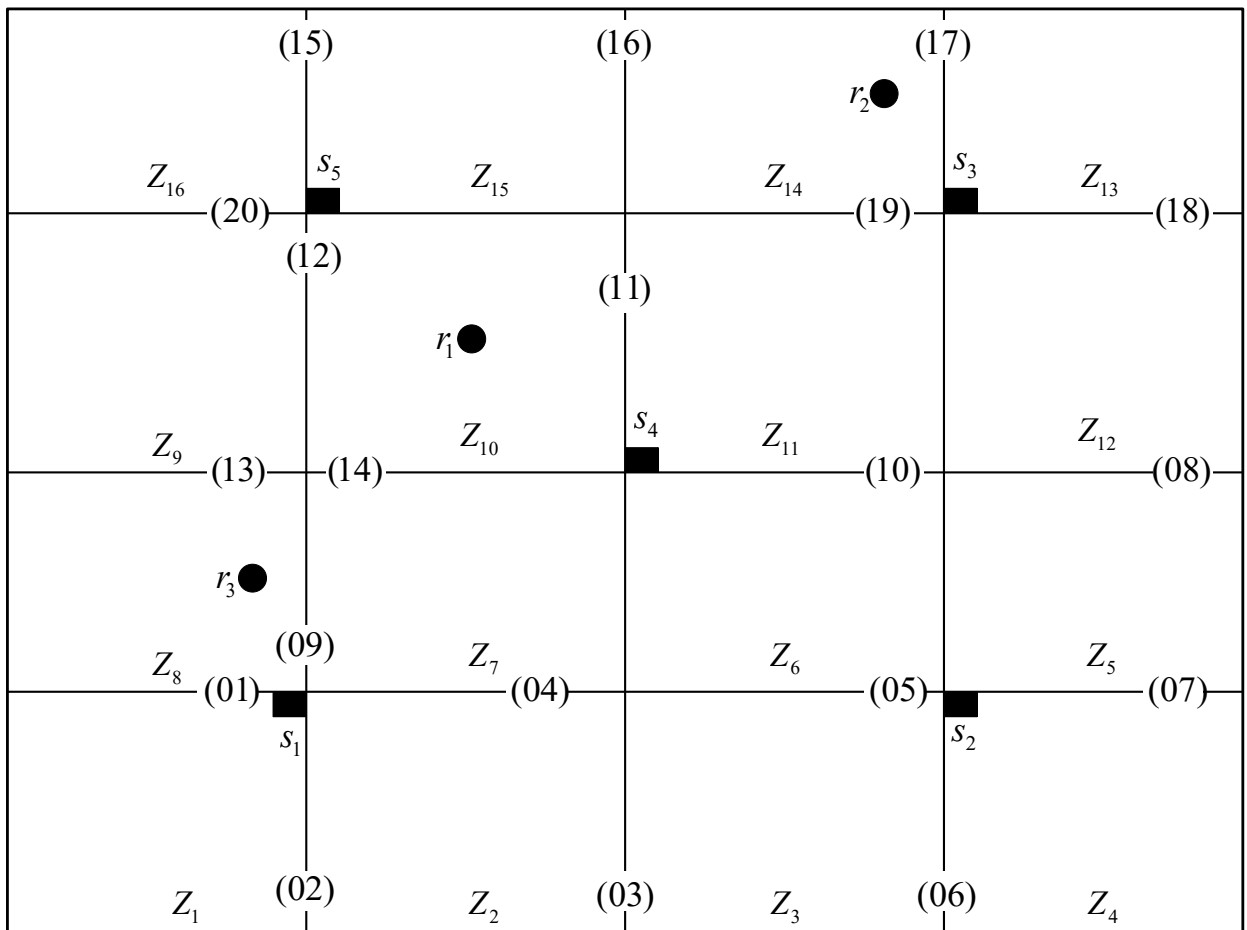

**Figure 7.** Service space of Simulation.

Five charging stations are deployed in the service space, and the location of each charging station is shown in Table 2. The charging stations can communicate with each other.

$$
\begin{aligned}
\Omega = \{ & Z_1 = (0 \leq x < 10, 0 \leq y < 10), Z_2 = (10 \leq x < 20, 0 \leq y < 10), \\
& Z_3 = (20 \leq x < 30, 0 \leq y < 10), Z_4 = (30 \leq x \leq 40, 0 \leq y < 10), \\
& Z_5 = (30 \leq x \leq 40, 10 \leq y < 20), Z_6 = (20 \leq x < 30, 10 \leq y < 20), \\
& Z_7 = (10 \leq x < 20, 10 \leq y < 20), Z_8 = (0 \leq x < 10, 10 \leq y < 20), \\
& Z_9 = (0 \leq x < 10, 20 \leq y < 30), Z_{10} = (10 \leq x < 20, 20 \leq y < 30), \\
& Z_{11} = (20 \leq x < 30, 20 \leq y < 30), Z_{12} = (30 \leq x \leq 40, 20 \leq y < 30), \\
& Z_{13} = (30 \leq x \leq 40, 30 \leq y \leq 40), Z_{14} = (20 \leq x < 30, 30 \leq y \leq 40), \\
& Z_{15} = (10 \leq x < 20, 30 \leq y \leq 40), Z_{16} = (0 \leq x < 10, 30 \leq y \leq 40) \}.
\end{aligned}
\tag{12}
$$

**Table 2.** Coordinates of deployed charging stations.

| Charging Station | $s_1$ | $s_2$ | $s_3$ | $s_4$ | $s_5$ |
|---|---|---|---|---|---|
| Coordinates | (9, 9) | (31, 11) | (31, 31) | (21, 21) | (11, 31) |

$\Omega$ as shown in Formula (12).

There are 16 rooms in the service space, which are connected with each other. The influence degree of the room is a set of random values, as shown in Table 3.

**Table 3.** Influence degree of each room.

| Weight | Value | Weight | Value |
|---|---|---|---|
| $w_{Z_1}$ | 0.78 | $w_{Z_9}$ | 0.60 |
| $w_{Z_2}$ | 0.21 | $w_{Z_{10}}$ | 0.10 |
| $w_{Z_3}$ | 0.35 | $w_{Z_{11}}$ | 0.10 |
| $w_{Z_4}$ | 0.98 | $w_{Z_{12}}$ | 0.74 |
| $w_{Z_5}$ | 0.70 | $w_{Z_{13}}$ | 0.30 |
| $w_{Z_6}$ | 0.11 | $w_{Z_{14}}$ | 0.58 |
| $w_{Z_7}$ | 0.55 | $w_{Z_{15}}$ | 0.99 |
| $w_{Z_8}$ | 0.89 | $w_{Z_{16}}$ | 0.96 |

Six robots are deployed, and the task weight of each robot is shown in Table 4.

**Table 4.** Weight of each robot task.

| Weight | Value |
|---|---|
| $w_{r_1}$ | 0.78 |
| $w_{r_2}$ | 0.10 |
| $w_{r_3}$ | 0.92 |
| $w_{r_4}$ | 0.20 |
| $w_{r_5}$ | 0.75 |
| $w_{r_6}$ | 0.4 |

The queue $\Upsilon$ of robots waiting to be towed is set as shown in Table 5.

**Table 5.** Coordinates of robots waiting to be towed.

| Robot | Coordinates |
|---|---|
| $r_2$ | (28, 36) |
| $r_3$ | (7, 15) |
| $r_1$ | (15, 25) |

The queue $\Psi$ of idle and available charging stations is shown in Table 6.

**Table 6.** Coordinates of idle and available charging stations.

| Robot | Coordinates |
|---|---|
| $s_2$ | (31, 11) |
| $s_3$ | (31, 31) |
| $s_4$ | (21, 21) |

All robot-trailers are set to drive at a uniform speed, and the driving speed is set to 1 m/s. $\alpha$, $\beta$ and $\phi$ are adjustment coefficients which values are set to 1. In this Simulation, we compare the DTNS proposed in this paper with the FCFS scheduling. In practice, it takes time for robot-trailer to load and unload robots. Because we are concerned with the comparison of scheduling efficiency, the time of robot-trailer loading and unloading robots in DTNS and FCFS scheduling algorithms is the same, so the time comparison in this experiment is not included in the time of robot-trailer loading and unloading robots. We

discuss three conditions of two scenarios. The first scenario is that when the robot-trailer plans the path, no other robot sends a charging request. The second scenario is that when the robot-trailer plans the path, other robots send the request to be towed for recharging.

Scenario 1: we first discuss the efficiency of the scheduling algorithm when there is only one robot-trailer. If the initial position of the robot-trailer $l_1$ is at $s_1$, the charging path obtained by using FCFS and DTNS is shown in Table 7.

**Table 7.** Paths obtained by FCFS and DTNS under the condition of one robot-trailer in scenario 1.

| Robot-Trailer | FCFS | DTNS |
|---|---|---|
| $l_1$ | $s_1 \rightarrow (1) \rightarrow (9) \rightarrow (14) \rightarrow (11) \rightarrow (19)$ $\rightarrow r_2 \rightarrow (17) \rightarrow (18) \rightarrow (8) \rightarrow (1) \rightarrow (7)$ $\rightarrow s_2 \rightarrow (6) \rightarrow (3) \rightarrow (4) \rightarrow (9) \rightarrow r_3$ $\rightarrow (9) \rightarrow (14) \rightarrow (11) \rightarrow (19) \rightarrow (17)$ $\rightarrow s_3 \rightarrow (17) \rightarrow (19) \rightarrow (11) \rightarrow r_1$ $\rightarrow (11) \rightarrow s_4$ | $s_1 \rightarrow (1) \rightarrow r_3 \rightarrow (9) \rightarrow (14) \rightarrow (11)$ $\rightarrow s_4 \rightarrow (11) \rightarrow r_1 \rightarrow (11) \rightarrow s_3 \rightarrow (17)$ $\rightarrow r_2 \rightarrow (17) \rightarrow (18) \rightarrow (8) \rightarrow (7) \rightarrow s_2$ |

The towing service time of each requesting robot and the delay expectation of the whole robot service system are shown in Figure 8, where $t_{tra}(r_i)$ represents the service time of towing robot $r_i$ to charge. For the FCFS scheduling, the delay time of robots $r_1$, $r_2$ and $r_3$ is 81.1317 s, 167.2635 s and 209.2815 s respectively. For the DTNS scheduling, the delay time of robots $r_1$, $r_2$ and $r_3$ is 35.8998 s, 77.9178s and 132.3038 s respectively. The delay expectation of the whole robot service system is 542.0944 for FCFS algorithm and 223.5127 for DTNS algorithm.

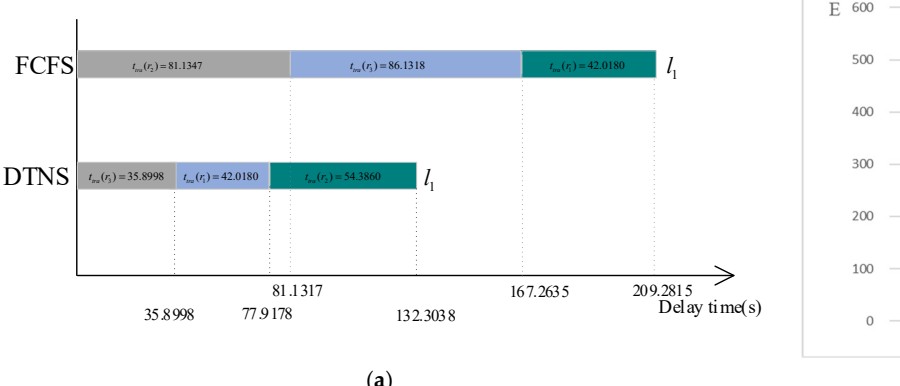
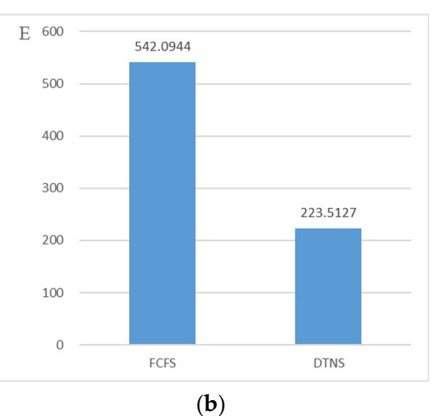

(**a**)                                              (**b**)

**Figure 8.** Comparison of efficiency under the condition of one robot-trailer in scenario 1. (**a**) Comparison of towing service time. (**b**) Comparison of delay expectations.

Under the condition of two robot trailers, let the initial position of the robot-trailer $l_1$ be at $s_1$ and the initial position of the robot-trailer $l_2$ be at $s_5$. Suppose $l_1$ gets the data a little earlier than $l_2$. The charging paths obtained by using the two scheduling algorithms are shown in Table 8.

**Table 8.** Paths obtained by FCFS and DTNS under the condition of two robot-trailer in scenario 1.

| Robot-Trailer | FCFS | DTNS |
|---|---|---|
| $l_1$ | $s_1 \rightarrow (1) \rightarrow (9) \rightarrow (14) \rightarrow (11) \rightarrow (19) \rightarrow r_2$ $\rightarrow (17) \rightarrow (18) \rightarrow (8) \rightarrow (7) \rightarrow s_2 \rightarrow (6) \rightarrow (5)$ $\rightarrow (10) \rightarrow (11) \rightarrow r_1 \rightarrow (1) \rightarrow s_4$ | $s_1 \rightarrow (1) \rightarrow r_3 \rightarrow (9) \rightarrow (14) \rightarrow s_4$ |
| $l_2$ | $s_5 \rightarrow (15) \rightarrow (20) \rightarrow (13) \rightarrow r_3 \rightarrow (9) \rightarrow (14)$ $\rightarrow (11) \rightarrow (19) \rightarrow (17) \rightarrow s_3$ | $s_5 \rightarrow (16) \rightarrow r_2 \rightarrow (17) \rightarrow s_3$ $\rightarrow (17) \rightarrow (19) \rightarrow (11) \rightarrow r_1$ $\rightarrow (11) \rightarrow (10) \rightarrow (5) \rightarrow (6) \rightarrow s_2$ |

The towing charging service time of each requesting robot and the delay expectation of the whole robot service system are shown in Figure 9. For the FCFS algorithm, the delay time of robots $r_1$, $r_2$ and $r_3$ is 135.9195 s, 81.1347 s and 84.2742 s respectively. For the

DTNS algorithm, the delay time of robots $r_1$, $r_2$ and $r_3$ is 102.141 s, 28.2741 s and 35.8998 s respectively. The delay expectation of the whole robot service system is 327.3172 for FCFS algorithm and 196.4885 for DTNS algorithm.

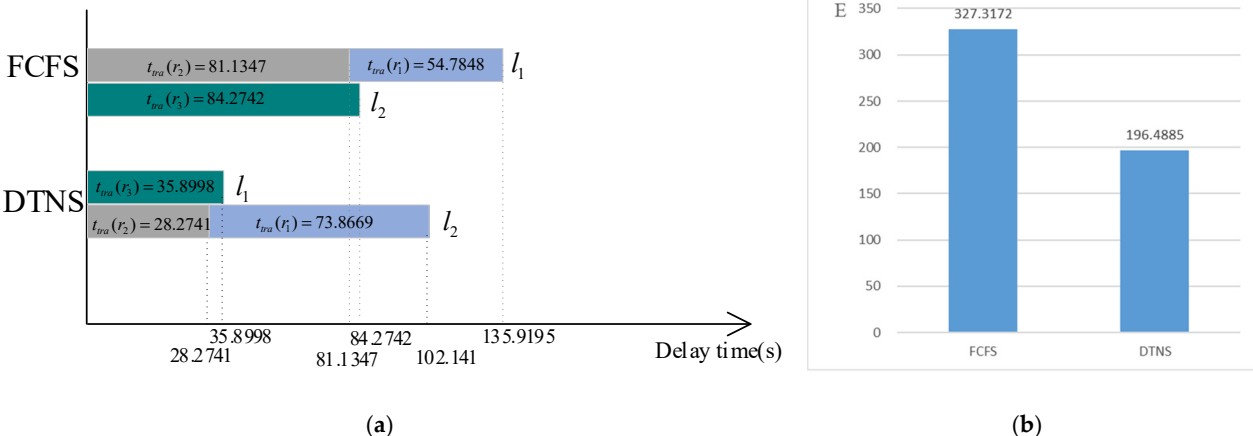

(**a**)

(**b**)

**Figure 9.** Comparison of efficiency under the condition of two robot-trailer in scenario 1. (**a**) Comparison of towing service time. (**b**) Comparison of delay expectations.

Under the condition of three robot-trailers, the initial position of the robot-trailer $l_1$ is set at $s_1$, the initial position of the robot-trailer $l_2$ is set at $s_5$, and the initial position of the robot-trailer $l_3$ is set at $s_4$. The charging path obtained is shown in Table 9.

**Table 9.** Paths obtained by the two algorithms under the condition of three robot-trailer in scenario 1.

| Robot-Trailer | FCFS | DTNS |
|:---:|:---:|:---:|
| $l_1$ | $s_1 \rightarrow (1) \rightarrow (9) \rightarrow (14) \rightarrow (11) \rightarrow (19)$ $\rightarrow r_2 \rightarrow (17) \rightarrow (18) \rightarrow (8) \rightarrow (7) \rightarrow s_2$ | $s_1 \rightarrow (1) \rightarrow r_3 \rightarrow (9) \rightarrow$ $(14) \rightarrow (11) \rightarrow s_4$ |
| $l_2$ | $s_5 \rightarrow (15) \rightarrow (20) \rightarrow (13) \rightarrow r_3 \rightarrow (9)$ $\rightarrow (14) \rightarrow (11) \rightarrow (19) \rightarrow (17) \rightarrow s_3$ | $s_5 \rightarrow (16) \rightarrow r_2 \rightarrow (17) \rightarrow s_3$ |
| $l_3$ | $s_4 \rightarrow (11) \rightarrow r_1 \rightarrow (11) \rightarrow s_4$ | $s_4 \rightarrow (11) \rightarrow r_1 \rightarrow (11) \rightarrow (10)$ $\rightarrow (5) \rightarrow (6) \rightarrow s_2$ |

The towing charging service time of each requesting robot and the delay expectation of the whole robot service system are shown in Figure 10. For the FCFS algorithm, the delay time of robots $r_1$, $r_2$ and $r_3$ is 22.9359 s, 81.1347 s and 84.2742 s respectively. For the DTNS algorithm, the delay time of robots $r_1$, $r_2$ and $r_3$ is 54.7848 s, 28.2741 s and 35.8998 s respectively. The delay expectation of the whole robot service system is 227.8915 for FCFS algorithm and 132.4156 for DTNS algorithm.

As can be seen from Figures 8b, 9b and 10b, the delay expectation of the proposed DTNS is lower than that of FCFS, with a difference of nearly half. As the number of robot-trailers increases, the delay expectation of the two models decreases. As can be seen from Figures 8a, 9a and 10a, the delay time of the proposed DTNS is less than that of FCFS. Tables 7–9 show that most of the paths obtained by DTNS are shorter than those obtained by FCFS.

Scenario 2: It is assumed that during the execution of scheduling planning, a robot sends a towing recharging request. Suppose that after towing the robot in the dataset $\Upsilon$ for 30 s, $r_5$ sent a towed charging request and entered dataset $\Upsilon$. At the same time, the charging station $s_5$ has a robot fully charged to leave, which is idle and available, and its coordinates are stored in the dataset $\Psi$. The model of this work is low cache path planning, which can be processed immediately, while the first come first service model will be processed after all three robots are charged. Similarly, this experiment is divided into three conditions for discussion.

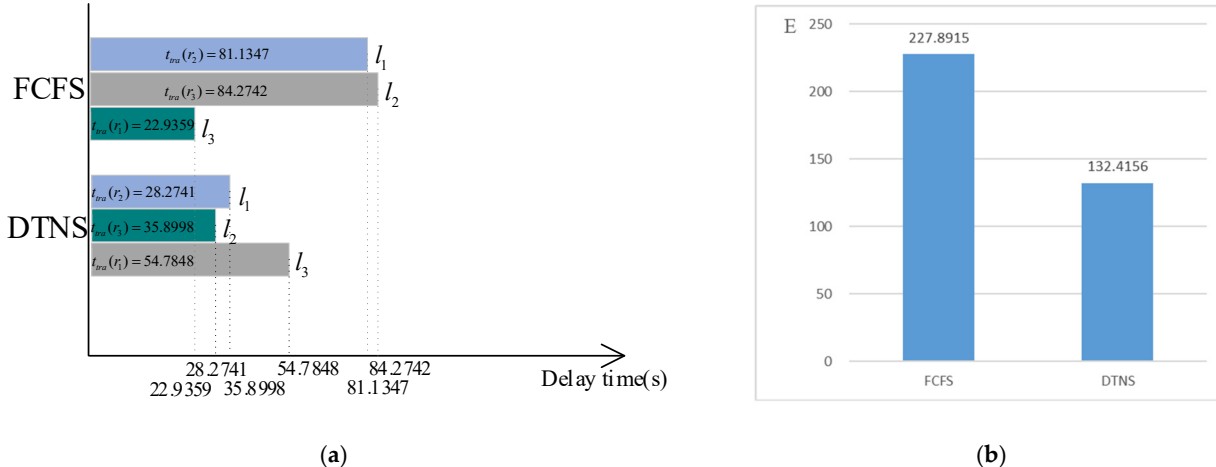

**Figure 10.** Comparison of efficiency under the condition of three robot-trailer in scenario 1. (**a**) Comparison of towing service time. (**b**) Comparison of delay expectations.

Under the condition of one robot trailers, the initial position of the robot-trailer is set at $s_1$. Using two scheduling models, the charging paths are shown in Table 10.

**Table 10.** Paths obtained by FCFS and DTNS under the condition of one robot-trailer in scenario 2.

| Robot-Trailer | FCFS | DTNS |
|---|---|---|
| $l_1$ | $s_1 \rightarrow (1) \rightarrow (9) \rightarrow (14) \rightarrow (11) \rightarrow (19) \rightarrow r_2 \rightarrow (17)$ $\rightarrow (18) \rightarrow (8) \rightarrow (7) \rightarrow s_2 \rightarrow (6) \rightarrow (3) \rightarrow (4) \rightarrow$ $(9) \rightarrow r_3 \rightarrow (9) \rightarrow (14) \rightarrow (11) \rightarrow (19) \rightarrow (17) \rightarrow$ $s_3 \rightarrow (17) \rightarrow (19) \rightarrow (11) \rightarrow r_1 \rightarrow (11) \rightarrow s_4 \rightarrow (10)$ $\rightarrow (5) \rightarrow (6) \rightarrow (7) \rightarrow r_5 \rightarrow (8) \rightarrow (17) \rightarrow (16) \rightarrow s_5$ | $s_1 \rightarrow (1) \rightarrow r_3 \rightarrow (9) \rightarrow (14) \rightarrow$ $(11) \rightarrow s_4 \rightarrow (11) \rightarrow r_1 \rightarrow (11) \rightarrow$ $(19) \rightarrow (17) \rightarrow s_3 \rightarrow (17) \rightarrow r_2 \rightarrow$ $(16) \rightarrow s_5 \rightarrow (16) \rightarrow (17) \rightarrow (18) \rightarrow$ $(8) \rightarrow r_5 \rightarrow (7) \rightarrow s_2$ |

The towing charging service time of each requesting robot and the delay expectation of the whole robot service system are shown in Figure 11. For the FCFS algorithm, the delay time of robots $r_1$, $r_2$ and $r_3$ is 209.2815 s, 81.1347 s and 167.2635 s respectively. The delay time for robot $r_5$ is $81.1317 + 86.1318 + 42.0180 + 89.6028 - 30 = 268.8843$ s. For the DTNS algorithm, the delay time of robots $r_1$, $r_2$ and $r_3$ is 77.9178 s, 106.1919 s and 35.8998 s respectively. The delay time for robot $r_5$ is $35.8998 + 42.0180 + 28.2741 + 60.0429 - 30 = 136.2348$ s. The delay expectation of the whole robot service system is 871.055 for FCFS algorithm and 446.7986 for DTNS algorithm.

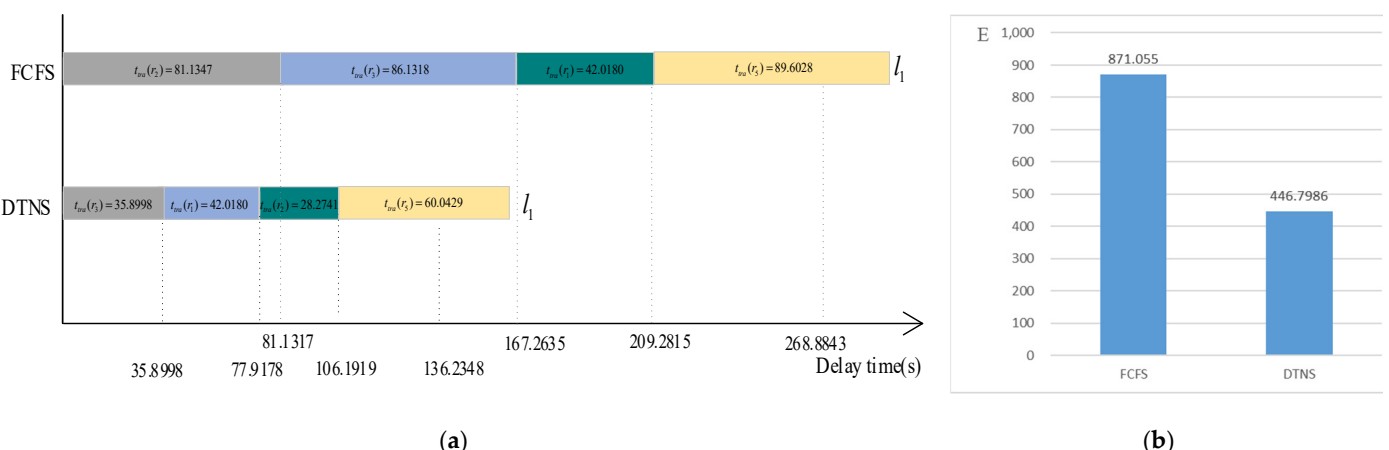

**Figure 11.** Comparison of efficiency under the condition of one robot-trailer in scenario 2. (**a**) Comparison of towing service time. (**b**) Comparison of delay expectations.

Comparing Figure 8a with Figure 11a, it is found that due to the entry of $r_5$ and $s_5$, the delay time of the robot $r_2$ is less, because it is sent to a charging station $s_5$ closer to it. The first come first service model needs to wait until it is all run before dealing with the following $r_5$ and $s_5$.

Under the condition of two robot trailers, let the initial position of the robot-trailer $l_1$ be at $s_1$ and the initial position of the robot-trailer $l_2$ be at $s_5$. The charging paths obtained by using the two scheduling models are shown in Table 11.

**Table 11.** Paths obtained by FCFS and DTNS under the condition of two robot-trailer in scenario 2.

| Robot-Trailer | FCFS | DTNS |
|---|---|---|
| $l_1$ | $s_1 \to (1) \to (9) \to (14) \to (11) \to (19) \to r_2$ $\to (17) \to (18) \to (8) \to (7) \to s_2 \to (6)$ $\to (5) \to (10) \to (11) \to r1 \to (11) \to s_4$ $\to (10) \to (5) \to (6) \to (7) \to r_5 \to (8)$ $\to (18) \to (17) \to (16) \to s_2$ | $s_1 \to (1) \to r_3 \to (9) \to (14) \to (11)$ $\to s_4 \to (10) \to (5) \to (6) \to (7)$ $\to r_5$ |
| $l_2$ | $s_5 \to (15) \to (20) \to (13) \to r_3 \to (9)$ $\to (14) \to (11) \to (19) \to (17) \to s_3$ | $s_5 \to (16) \to r_2 \to (17) \to s_3 \to (17) \to (19) \to$ $(11) \to r_1 \to (11) \to (10) \to (5) \to (6) \to s_2$ |

The towing charging service time of each requesting robot and the delay expectation of the whole robot service system are shown in Figure 12. For the FCFS algorithm, the delay time of robots $r_1$, $r_2$ and $r_3$ is 235.9195 s, 81.1347 s and 84.2742 s respectively. The delay time for robot $r_5$ is 81.1317 + 54.7848 + 89.6028-30 = 160.7043 s. For the DTNS algorithm, the delay time of robots $r_1$, $r_2$ and $r_3$ is 102.141 s, 28.2741 s and 35.8998 s respectively. The delay time for robot $r_5$ is 35.8998 + 89.6028-30 = 95.5026 s. The delay expectation of the whole robot service system is 610.8246 for FCFS algorithm and 312.5677 for DTNS algorithm.

Under the condition of three robot-trailers, the initial position of the robot-trailer $l_1$ is set at $s_1$, the initial position of the robot-trailer $l_2$ is set at $s_5$, and the initial position of the robot-trailer $l_3$ is set at $s_4$. The charging path obtained is shown in Table 12.

The towing charging service time of each requesting robot and the delay expectation of the whole robot service system are shown in Figure 11. For the FCFS, the delay time of robots $r_1$, $r_2$ and $r_3$ is 22.9359 s, 81.1347 s and 84.2742 s respectively. The delay time for robot $r_5$ is 84.2742 + 60.0429 - 30 = 114.3171 s. For the DTNS algorithm, the delay time of robots $r_1$, $r_2$ and $r_3$ is 54.7848 s, 28.2741 s and 35.8998 s respectively. The delay time for robot $r_5$ is 30 + 68.4886 = 98.4886 s. The time for the robot-trailer to finish serving the robot is 28.2741 s. Wait until 30 s, $r_5$ is stored in dataset $\Upsilon$. The delay expectation of the whole robot service system is 871.055 for FCFS algorithm and 446.7986 for DTNS algorithm.

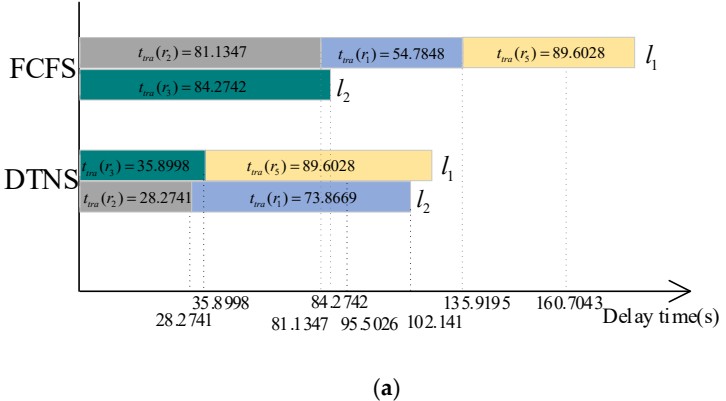
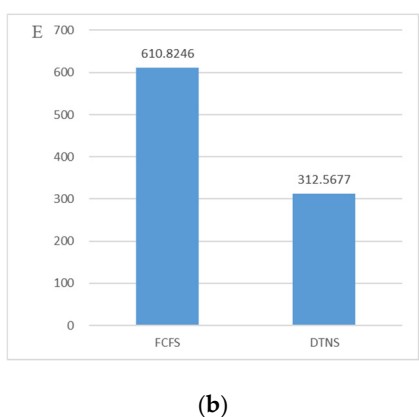

(a)                                                                                    (b)

**Figure 12.** Comparison of efficiency under the condition of two robot-trailer in scenario 2. (**a**) Comparison of towing service time. (**b**) Comparison of delay expectations.

**Table 12.** Paths obtained by FCFS and DTNS under the condition of three robot-trailer in scenario 2.

| Robot-Trailer | FCFS | DTNS |
|---|---|---|
| $l_1$ | $s_1 \rightarrow (1) \rightarrow (9) \rightarrow (14) \rightarrow (11) \rightarrow (19) \rightarrow r_2$ $\rightarrow (17) \rightarrow (18) \rightarrow (8) \rightarrow (7) \rightarrow s_2 \rightarrow (7)$ $\rightarrow r_5 \rightarrow (8) \rightarrow (18) \rightarrow (17) \rightarrow (16) \rightarrow s_5$ | $s_1 \rightarrow (1) \rightarrow r_3 \rightarrow (9) \rightarrow (14)$ $\rightarrow (11) \rightarrow s_4$ |
| $l_2$ | $s_5 \rightarrow (15) \rightarrow (20) \rightarrow (13) \rightarrow r_3 \rightarrow (9)$ $\rightarrow (14) \rightarrow (11) \rightarrow (19) \rightarrow (17) \rightarrow s_3$ | $s_5 \rightarrow (16) \rightarrow r_2 \rightarrow (17) \rightarrow s_3 \rightarrow (18) \rightarrow (8)$ $\rightarrow r_5 \rightarrow (8) \rightarrow (18) \rightarrow (17) \rightarrow (16) \rightarrow s_5$ |
| $l_3$ | $s_4 \rightarrow (11) \rightarrow r_1 \rightarrow (11) \rightarrow s_4$ | $s_4 \rightarrow (11) \rightarrow r_1 \rightarrow (11) \rightarrow$ $(10) \rightarrow (5) \rightarrow (6) \rightarrow s_2$ |

As can be seen from Figures 11a, 12a and 13a, the delay time of DTNS model is less than that of FCFS model. Tables 10–12 show that the most of paths obtained by most DTNS is shorter than that obtained by FCFS. For the FCFS scheduling model, it is necessary to wait until all three robots are serviced before serving the incoming robot $r_5$. However, after 30 s, the DTNS scheduling model can serve the incoming robot $r_5$ as long as there is free robot-trailer available. For the scenario 2, it can be seen from Figures 11b, 12b and 13b, because there are more robots serving, the overall service delay is higher than that of the first scenario, but the overall trend is decreasing with the increase of robot trailers. The delay expectation of DTNS scheduling model proposed in this paper is lower than that of FCFS.

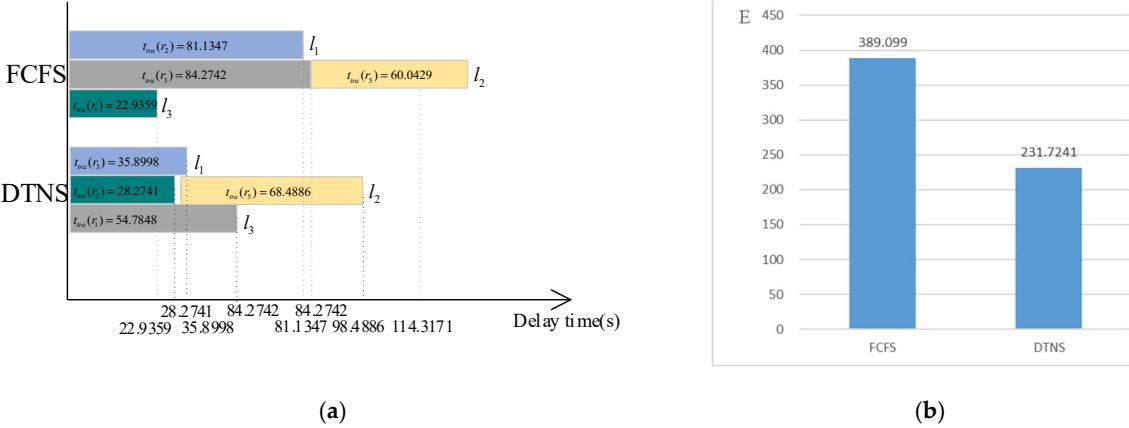

(**a**)                    (**b**)

**Figure 13.** Comparison of efficiency under the condition of three robot-trailer in scenario 2. (**a**) Comparison of towing service time. (**b**) Comparison of delay expectations.

## 6. Conclusions

When the mobile robot fails or the residual energy cannot make the robot reach the charging station, it is particularly important to adopt the automatic distributed scheduling of robot trailer for the power-off robot. It is also the requirement for human beings to liberate the labor force and move towards the intelligent era. The research in this paper is based on the charging station with communication function, and studies the optimization of the robot-trailer to complete the towing service that robot-trailer tow the robot to recharge.

The distributed scheduling model in this paper is based on the premise that the operating robot sends a towing request to the distributed charging station, whose communication success rate has a good value when the communication threshold is not high. When the charging station adopts the fixed position and random uniform distribution deployment mode, the complete communication success is achieved when the communication distance threshold is 15 m and 25 m respectively in the simulation.

The optimization model of towing scheduling studied in this paper takes into account the task weight of the robot and the impact of power failure on the room, which is of practical significance. The DTNS scheduling and two-step path planning algorithms

designed in this paper have lower delay expectations than other scheduling models. In the simulation experiment, the path generated by DTNS is also relatively short. DTNS has strong flexibility, and can timely handle the new request of robot towed to recharge and optimize the scheduling path. In the case of 1, 2 and 3 robot trailers in the service space, when there are some robots sending a towing request, the towing delay is reduced by 48.71%, 48.83% and 40.45%, and when no robot sends a towing request, the delay is reduced by 58.77%, 39.97% and 41.90%, compared with the FCFS algorithm. Therefore, the authors conclude that DTNS is an effective way to solve the charging efficiency of robot that can be towed to recharge by robot-trailer.

**Author Contributions:** Conceptualization, Y.D. and M.X.; methodology, Y.D.; validation, Y.D. and M.X.; formal analysis, T.L.; data curation, T.L.; writing—original draft preparation, Y.D. and M.X.; writing—review and editing, M.X. and S.Z.; visualization, S.Z.; project administration, M.X. All authors have read and agreed to the published version of the manuscript.

**Funding:** This work was financially supported in part by the National Natural Science Foundation of China under Grant 62266010, in part by the Cultivation Project of Guizhou University (No. [2019]57) and Research Project of Guizhou University for Talent Introduction (No. [2019]31).

**Institutional Review Board Statement:** Not applicable.

**Informed Consent Statement:** Not applicable.

**Data Availability Statement:** The measured data that support the findings of this study are simulated.

**Conflicts of Interest:** The authors declare that there is no conflict of interest regarding the publication of this paper.

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
