# Peer review of "Distributed Multi-Robot-Trailer Scheduling Based on Communication between Charging Stations for Robot Being Towed to Recharge"

_electronics, doi:10.3390/electronics12061402_

Round 1

Reviewer 1 Report

The authors investigate an optimization problem related to scheduling the recharging of robot-trailers, and propose a distributed scheduling method that utilizes a two-step path planner for the robot-trailers. The effectiveness of this approach is demonstrated through simulations conducted by the authors.

Strengths:

+ Identifying an optimal schedule for towing discharged robots to nearby charging stations is an interesting problem.

+ The two-step approach for path planning seems effective in terms of reducing the delay.

+ The proposed optimization formulation seems valid.

Weaknesses:

- The problem space is simplistic. Whether it can be easily extended for real-world scenarios is questionable.

- Details on simulation settings are missing. For example, what simulation tool was used? What model was used to model the robot motion? How is the communication between robots modeled in the simulation?

- It appears that the simulations conducted in the study may have been conducted in relatively simple environments, which may not fully capture the challenges of scheduling and recharging robot-trailers in more complex real-world settings.

- It is unclear how the proposed approach would perform in more realistic environments, as the study may not have adequately tested its performance under more complex or challenging operating conditions.

- It is not clear how the weights for robots were determined.

- The English language used in this paper could be improved significantly to enhance clarity and readability for readers.

Reviewer 2 Report

After reviewing this work, it should be noted that the manuscript was developed at a very high scientific level in terms of content and at a weaker level in the field of methodology (text continuity, editorial errors, formatting, style, etc.), thus contributing to the deepening of knowledge not only concerning the area of electrical power supply, including in its fields (electronics, electrical engineering), but especially in the area of control, distribution and navigation systems in relation to the management of industrial mobile robots (electrical power automation, robotics, etc.). 

This is evidenced, among other things, by considered by the authors of this study both a critical analysis of the literature on the subject of research [1]-[30], and above all, the process of analysis and research inquiries of distributed scheduling of multi-robot-trailers for communication between charging stations for a robot towed for charging, which is distributed scheduling model created by the authors, in the form of DTNS (Distributed Three Nodes Service) based on communication between charging stations for the purpose of reducing the delay of towing a robot, such as in relation to the method using FCFS (First Come First Serve) scheduling. 

In the research aspect, the paper is interesting, presenting an innovative idea/technical approach to solving the research problem related to the research process, concerning the considerations associated with the proposed distributed scheduling of multi-robot-trailers based on communication between charging stations for a robot towed for charging using the DTNS technique, on the basis of the two-steps path-planner algorithm designed by the authors of this paper based on decision factor and travel path. 

The new contribution of the work is the technical approach, the methods used, including the research carried out and their in-depth analysis, based on a detailed consideration of the advantages and disadvantages of existing solutions (models, methods, etc.), especially in terms of achieving a lower delay expectation than other scheduling models. 

In confirming the above assessment, it should be noted that the authors of this work, in addition to graphical presentation, discussion and proper analysis of their research and simulation tests in the aspect of the scientific problem that concerns them, highlighted their research inquiries both through graphical depiction in the form of diagrams, scheduling algorithms, etc. (Figures 1-4), tabular summaries (Tables 1-12), and, most importantly, the simulations they performed (Figures 5-13), with detailed ongoing commentary and in-depth analysis of the results they obtained, which allowed them to formulate important insights and final conclusions that are reflected in practical applications. 

In addition, both in the methodological aspect (structure, layout of the work), in exception to minor inaccuracies cited in this review, and mainly in the content aspect, based on the aptly conducted analysis of the current state of knowledge in the consideration of existing solutions for research in the application of the technique of distributed trailer scheduling with multiple robots based on communication between charging stations for a robot towed for charging, the authors have made the necessary research, testing and simulation experiments at a very high (professional) scientific level. 

This is very important from the point of view of existing knowledge, as evidenced by the research results obtained by the authors in terms of the strategy adopted to solve the research problem. 

While reviewing this article, except for minor editing errors and some inaccuracies occurring in the abstract section and in the conclusion of the reviewed work in the methodological aspect, I did not find any other shortcomings having a key impact on the level and quality of this study. 

Abstract 

In accordance with the recommendations of reputable publishing houses and journals, e.g. IEEE TTE, IEEE Access, Wiley and Sons, or MDPI, part of the abstract should contain the following main elements: introduction (reference to the subject matter of the study), clear definition of the aim of the work, approximation/addressing of the potential solution to the problem/methods, and response to on the basis of the research, test, experience, developed mathematical model, to formulate relevant observations and final conclusions. 

The abstract should not exceed 200 words, and in this article there are 229 of them. In addition to inaccuracies such as the explanation of abbreviations, including their duplication, in my opinion, the abstract section of this paper mainly lacks a clear statement of the purpose of the article 

Inaccuracies observed in part of the abstract of this manuscript:

  1. No explicit statement of the purpose of the research in this manuscript. 
  2. In the abstract, it is not recommended to explain abbreviations and designations as it was done in this article for the abbreviations DTNS, FCFS. 
  3. There is a lack of clear reference to both the expected and predicted results of the research (analysis, model, simulations) in terms of the approach used/methods employed. 

The rest of the work 

Minor inaccuracies noted in the rest of the work: 

1. Referring to the literature references [1]-[30] presented in this paper, with the exception of duplication [1] on page 1, point 40, all other references were presented by the authors of this paper properly with unambiguous notation and in the correct order. However, except for references [28] on page 5, [29] on page 6 and [30] on page 12, the authors cited the remaining references [1]-[27] on the first four pages, and the paper contains 21 pages. So it can be thought that on the remaining pages of the manuscript (pp. 13-21), the authors did not need to use the literature references of the subject of the study, please explain this kind of approach. 

2. Duplication of explanations of some abbreviations both in the abstract and in the rest of the paper, e.g. DTNS and FCFS (in the abstract) and on pages 8 and 12 (DTNS) and on pages 12 and 14 (FCFS), respectively. In addition, I noticed a lack of clarity in their explanations, e.g., page 14, points 491-492 for DTNS and FCFS. Other inaccuracies regarding abbreviations in terms of unambiguous notation are, e.g., DPMS on p. 4, points 172 and 176, respectively, and the lack of explanation of some abbreviations, e.g., GPS, or GPS/IMU on p. 2 or GPASP on p. 1, point 44. Please review the entire work in this regard and make appropriate corrections. 

3. In terms of the cited references to figures and tables, I found no reference to Figure 4 in the text of the paper. 

4. Using short sequences in the work, e.g., p. 1, points 35 (6 words) and 40-41 (8 words), respectively, or p. 2, point 56 (5 words). Similarly, in contrast, the use of excessively long sequences, e.g., p. 2, points 66-71 (66 words), p. 3, points 110-115 (70 words), or points 124-129 (65 words), etc. Please check and correct the aforementioned inaccuracies accordingly. 

5. The content contained in the paper should be written in the impersonal form or in the 3rd person, not as presented in this article both in the abstract section and in the rest of the paper in terms of we ..., us ..., our ..., etc. For example, we proposed ..., we design ..., or our ... (abstract), and pp. 2-3, 6, 9, 11-12, 14 (for we ...); pp. 2, 8, 11-12, 17 (for our ...) and p. 9 for us ..., etc. (the rest of the work). Please check the entire work in this regard. 

6. The work observed both the use of text continuity, e.g., p. 2, points 54-74; p. 3, points 110-140; p. 3-4, points 141-193; p. 5, points 205-218; p. 7-8, points 310-325, and p. 9, points 384-400, as well as the repeated use of the word Aiming ..., e.g., p. 3, 8 times. Please check your entire work in this regard and make appropriate corrections. 

7. In the paper, I noticed an incorrect order of chapter numbering, namely duplication of chapter 3 numbering, so starting from page 6, item 269 to the end of the paper, the numbering of chapters and subchapters presented in the reviewed manuscript should be corrected. 

8. In the article, I observed several editing errors amounting to mistakes in terms of both the failure to keep the writing unambiguous and the use of punctuation marks. For example, page 2, point 50 for 4.; page 3, point 136 for be obiective ..., or point 140 for Pareto Archiving Evolutionary strategies (PAEs), etc. In addition, text formatting errors, such as in the case of the description for Figure 5 (p. 11, points 433-434), as well as the poor quality of Figure 3. Please review the entire article in detail in this regard and make appropriate corrections. 

9. Minor errors in the accompanying literature list of the subject of the study in terms of the use of punctuation marks, failure to keep the notation unambiguous, etc., such as items 2, 9, or 25. 

10. Despite the fact that the chapters on the research process, i.e., Chapter 4 (5) Simulation and analysis of this manuscript on the research results obtained, were prepared by the authors at a very high (professional) scientific level, taking into account the research results obtained, with a current commentary, as well as an in-depth and accurate analysis of the results obtained, in the conclusion of this work Chapter 5 (6), it is recommended to refer to the research results obtained in more detail, including by confirming the data obtained. In view of the above, I believe that the final conclusions are presented too generally. 

Strong aspects:

The technical approach/methods used (DTNS distributed scheduling models, FCFS, comparative analysis, etc.), idea and explanation of the problem, the in-depth analysis of the research results obtained, supported by mathematical analysis (1)-(12), graphical illustration and tabulation, and the formulation of valuable and at the same time accurate final conclusions, the relevance in terms of the methods used and the ability to use them.

Weak aspects:

Minor shortcomings that have no significant impact on the quality of the reviewed work, i.e. editing errors and poor quality of the methodological part of the abstract and conclusion

Recommended changes:

Regardless of the Editor's decision, at this stage of the work, I would recommend that the authors of this paper make the corrections contained in the above review (weak aspects), including points 1-3 (abstract) and 1-10 (rest of the paper).

Author Response

Dear reviewer,

        Please see the attachment, Thank you.

Reviewer 3 Report

The article shows the novel design of a distributed scheduling system of transportation vehicles, i.e. Distributed Three Nodes Service (DTNS) scheduling, which is used for robot trailer deployment systems. The authors demonstrated a clear advantage of reduced transportation time with DTNS, by 40% - 59%, when compared to the existing First Come First Served (FCFS) scheduling. They attribute this higher efficiency to the two-step path-planner which facilitates more stable communication and reduces system delay.

To my opinion, the results presented in this article is of high relevance to scientists working on electrical and autonomous vehicles. The development of a more effective algorithm for automated vehicle scheduling is crucial for their widespread implemention. I think the introduction is logically reasoned, and comprehensive in its summarization of previous work in this field.

Still, I find that this article needs some minor revision. Apart from the numerous grammatical mistakes and typos, there are some questions I have that I think should be considered, both in the design of experiments and in the discussion of results. I shall list below some portions of the article that I consider should be revised.

- The results of Simulation 1 are presented in Figures 5 and 6. However, in FIgure 5, the x-axis denoted as "Request Distance" can be confusing; I was only able to find out that it meant "Communication Threshold" from the captions of this figure. In Figure 6, I find that some critical parameters are missing from the caption, such as the distance threshold, dth. (It is 10 m in this case, per Line 426, but it would be much easier to read if this parameter is included in the caption.)

- Regarding Figure 6(a), "the direct impact of space area on the request failure rate is not great and has a certain volatility, because the position of requests is random, but the overall trend is that the failure rate increases with the increase of area." (Lines 440 - 442) My understanding of this statement is that the trend of "failure rate increases with the increase of area" is intuitive, but not necessarily well supported by the data in Figure 6(a) due to its large volatility. My guess is that this conclusion could be better justified if the data of a different dth parameter were used, such that the curve might observe less fluctuations or noise, and the monotony of the curve might more pronounced. I would suggest refining this figure to better support the statement.

- In Figure 6(b), the x-axis representing "Space Area" is ranged from 100 to 600 m2. I think it is important to emphasize that the service center is positioned at (5 m, 5 m) in the figure. Otherwise it would be difficult to understand why the failure rate is 0 when the space area is 100 m2, since it is probable that the distance between two points, within a square of 10 m by 10 m, exceed 10 m.

- In the scenarios of Simulation 2, the expectation of waiting times of robots from the DTNS algorithm is compared against those from the FCFS algorithm. Given the coordinates of robots, robot trailers and service centers, I have no doubt about the precision of the waiting times, however I am skeptical about the accuracy of the numbers in the abstract, e.g. 40.45% or 58.77%. I think that the reduction in time could largely depend on the service space of experiments, i.e. spatial distribution of moving objects and doors. May I know if there have been another example of a service space, different from the layout described in Figure 7, used in these experiments to replicate the results? Would the magnitudes of time cost reduction still be at the same level, or higher or lower?

- This question might be out of context but given the large reduction of waiting times when DTNS is used instead of FCFS, I am curious whether it would be efficiency-wise viable to reduce the number of robot trailers in the map? How would the trade-off between the number of operating robot trailers and total waiting time of robots play out? Or would the option of cutting down trailers only be favorable when waiting times can be further reduced?

In summary, to my opinion the impact of the results obtained in this article is fit to be published in Electronics, but a minor revision where some questions are addressed is necessary.

Author Response

Dear reviewer,

Round 2

Reviewer 1 Report

My comments have been addressed satisfactorily.